# Diagnostic and Therapeutic Applications of Extracellular Vesicles in Interstitial Lung Diseases

**DOI:** 10.3390/diagnostics11010087

**Published:** 2021-01-07

**Authors:** Abdulrahman Ibrahim, Ahmed Ibrahim, Tanyalak Parimon

**Affiliations:** 1Faculty of Medicine, University of Queensland/Ochsner Clinical School, New Orleans, LA 70121, USA; v-abdibrahim@ochsner.org; 2Smidt Heart Institute, Cedars-Sinai Medical Center, Los Angeles, CA 90048, USA; ahmed.ibrahim@cshs.org; 3Pulmonary and Critical Care Division, Women’s Guild Lung Institute, Department of Medicine, Cedars-Sinai Medical Center, Los Angeles, CA 90048, USA

**Keywords:** extracellular vesicles, exosomes, pathogenesis, biomarkers, therapeutic capability, interstitial lung diseases, lung fibrosis

## Abstract

Interstitial lung diseases (ILDs) are chronic irreversible pulmonary conditions with significant morbidity and mortality. Diagnostic approaches to ILDs are complex and multifactorial. Effective therapeutic interventions are continuously investigated and explored with substantial progress, thanks to advances in basic understanding and translational efforts. Extracellular vesicles (EVs) offer a new paradigm in diagnosis and treatment. This leads to two significant implications: new disease biomarker discovery that enables reliable diagnosis and disease assessment and the development of regenerative medicine therapeutics that target fibroproliferative processes in diseased lung tissue. In this review, we discuss the current understanding of the role of diseased tissue-derived EVs in the development of interstitial lung diseases, the utility of these EVs as diagnostic and prognostic tools, and the existing therapeutic utility of EVs. Furthermore, we review the potential therapeutic application of EVs derived from various cellular sources.

## 1. Introduction

Interstitial lung diseases (ILDs) are a wide spectrum of diffuse parenchymal pulmonary conditions indicated by inflammatory changes in the alveoli. ILDs may present either idiopathically or by a sequela to preexisting comorbidities such as connective tissue, autoimmune diseases, or secondary to biological, chemical, or fine particle exposure [1,2]. The American Thoracic Society (ATS) and European Respiratory Society (ERS) designated the term idiopathic interstitial pneumonia (IIP) for ILDs of unknown etiologies [3]. IIPs are further subdivided as major, rare, and unclassified IIP [4,5,6]. The natural course of ILDs is characterized by chronic, progressive, and irreversible fibrosis with significant morbidity and mortality [6,7,8,9]. While the management of non-idiopathic ILDs relies on addressing the underlying cause of disease, the standard treatment approaches for IIP include antifibrotic therapy or lung transplantation [10,11,12,13,14]. Although current treatments provide significant morbidity and mortality benefits [15], they are not curative. Extensive research in the pathogenesis of lung fibroproliferation has opened new avenues of therapeutic intervention. ILDs continue to levy significant burdens on morbidity, mortality, and healthcare expenditure worldwide [16,17,18,19]. Further understanding of the underlying pathogenic mechanisms is essential for more effective antifibrotic treatment approaches. A new and promising avenue of investigation relates to the role of extracellular vesicles (EVs) in parenchymal lung injury and interstitial fibrosis. The role of EVs in the pathogenesis of nonmalignant chronic respiratory conditions has been extensively described in ILDs [20,21,22], asthma [23,24], chronic obstructive pulmonary diseases (COPD) [23,25,26,27], and pulmonary hypertension [28,29]. These studies provide an evidence base for the clinical translation of EVs in ILD diagnosis and the utilization of therapeutic EVs for treatment.

EVs are easily quantifiable and have features amenable to direct (i.e., as treatments) and indirect (i.e., as diagnostic tools) therapeutic applications. These include stability, cargo transfer, and direct regulation of disease pathogenesis. The current gold standard of ILDs diagnosis is invasive procedures such as transbronchial cryobiopsy through bronchoscopy or surgical lung biopsy [30]. These approaches are associated with an increased risk of procedural complications and clinical intolerability, particularly in patients with respiratory insufficiency [30,31,32]. As a result, noninvasive and reliable biomarkers for ILDs are needed for more effective clinical management and care. Additionally, EVs can also be used to monitor disease progression and response to treatment. The physiologic and pathogenic roles of EVs suggests their utility as a therapeutic platform in the restoration of organ homeostasis and reversal of tissue damage. Indeed, the steady migration of cells to EV therapy in regenerative medicine is primarily due to the versatility of these particles over traditional cell transplantation therapy [22,33,34,35]. The advancement of EV cargo content engineering renders EVs more versatile and customizable to several disease types [36]. Despite significant progress, some limitations have limited the transition of EVs to clinical studies [37]. Therefore, this review focuses on the current understanding of the molecular pathways that drive ILDs and the role of EVs in disease progression. We further discuss the applicability of EV translation in the context of biomarker development and EV therapy.

## 2. Extracellular Vesicles

Extracellular vesicles are lipid bilayer nanoparticles secreted by nearly all cell types and comprise several classes of particles delineated by size and pathway of biogenesis. For instance, exosomes are smaller (30–150 nm) particles produced through the late endosome with selective packaging of cargo. Ectosomes (e.g., microvesicles, apoptosomes, etc.) (100 nm–1 µm) are passively shed from the plasma membrane. However, evidence from the cancer biology and hypersensitivity models demonstrates that ectosome content will change based on different cell stimuli (i.e., stress, antigen exposure, etc.) [38]. Whether this is a reflection of the internal contents of the cell (in response to stimuli) or evidence of bona fide cargo-sorting remains to be verified. However, a recent report indicated that pyruvate kinase M2 (PKM2) can be sorted into hepatocellular carcinoma ectosomes through the sumoylation process [39]. EVs also carry a plethora of signaling mediators including several classes of non-coding RNAs (micro-RNAs, long noncoding RNAs, Piwi RNAs, etc.), proteins (growth factors, transcription factors, etc.), and lipids [40,41]. The discovery of extracellular vesicles [42,43] and their biological roles appears to be significant in a variety of human diseases, namely in chronic conditions such as pulmonary disease, cancer, diabetes, heart diseases, Alzheimer’s disease, kidney failure, liver cirrhosis, etc. [44,45,46,47,48,49]. Primarily, the demonstration of EVs facilitating intercellular communication has created a paradigm shift in our understanding of disease pathogenesis [50]. It offers novel tools that aid in the fundamental understanding of basic disease pathogenesis and the development of novel diagnostic and therapeutic platforms [51]. EVs play essential functional roles in maintaining tissue/organ homeostasis during normal development [52,53] and also in disease progression including tissue damage, fibrosis, and metastasis [54,55,56]. The biological functions of EVs rely on signal transduction of EV-associated cargo molecules which alter the transcriptome and epigenome of recipient cells. These EVs mediate communication and phenotypic change within and between tissues [45,46,47,48].

## 3. EVs in Ild Pathogenesis

The key cellular players in ILD pathogenesis include alveolar epithelial cells, lung fibroblasts, leukocytes, and endothelial cells [57]. Crosstalk between these cell types in the context of the injured lung microenvironment is mediated, at least in part, by EVs [58]. For instance, alveolar macrophage-, neutrophil-, and epithelial cell-derived EVs sampled from bronchoalveolar lavage fluid (BALF) mediate the acute and resolution phases of acute lung injury through the transfer of proinflammatory cytokines including TNF-α, IL-6, and IL1-β [59,60,61]. Indeed, in EVs isolated from human healthy volunteers, BALF carries MHC I and II, CD54, CD63, and the co-stimulatory molecule CD86, implicating their potential roles in immune regulation [62]. Similar studies in other chronic respiratory conditions (e.g., asthma, COPD, and lung cancer) confirm the disease-propagating role of EVs [63,64]. These findings highlight the importance of EVs in lung microenvironment signaling. Furthermore, through universal inflammatory and fibrotic processes, EVs are likely to play similarly critical roles in ILDs. 

### 3.1. EVs in Nonidiopathic Interstitial Lung Diseases

Nonidiopathic ILDs are defined as ILDs with known causes, such as connective tissue disorders, toxic environmental exposures, or chronic inflammatory lung diseases such as sarcoidosis [65]. The implication of EVs in non idiopathic ILDS is gaining increasing interest. A higher quantity of EV-associated proteins, called tissue factor (TF), were found in microvesicles (MVs, size ranged 0.05–1 μm) in the BALF of pulmonary fibrosis cases (4 known cause ILDs and 15 IIP patients). EV-bound tissue factor (TF) activity was associated with disease severity, highlighting the possible causal relationship between pathogenic MVs and lung damage [66]. TF-bearing MVs stimulated reactive oxygen species (ROS) production in human lung epithelial cell lines (A549 and NHBEC), suggesting that these MVs propagate injury (and fibrosis development) across the pulmonary epithelium.

A few studies of pulmonary sarcoidosis, a chronic inflammatory ILD, showed that sarcoidosis patients had a higher EV-burden in their BALF [67,68,69]. These EVs retained proinflammatory properties as demonstrated by their capacity to induce interferon-γ (IFNγ) and interleukin IL-13 production in peripheral blood mononuclear cells (PBMCs) and IL-8 from epithelial cells [67]. Others also reported that BALF-derived EVs from sarcoidosis patients stimulated monocytes to release IL-1β, IL-6, CCL2, and Tumor Necrosis Factor (TNF) [69]. These proinflammatory mediators promote lung inflammation that leads to fibrosis progression in patients with sarcoidosis. Therefore, EVs carry specific cargoes that can induce inflammatory responses in both immune and lung epithelial cells. Further characterization of these cargoes and their involvement in lung fibroproliferation is needed to understand the contribution of EVs to ILD pathogenesis and prognosis.

### 3.2. EVs in Idiopathic Interstitial Lung Diseases

The most common interstitial lung disease in this category is idiopathic pulmonary fibrosis (IPF). Single-cell RNA-sequencing of human and mouse lungs indicated that many cell types in the lungs are involved in IPF pathogenesis [70,71]. Most of those cells secrete EVs that drive lung fibroproliferative processes through activation of profibrotic signaling pathways such as transforming growth factor β (TGFβ) signaling, wingless N-type (Wnt)/β-catenin signaling, and cellular senescence.

EVs mediate lung fibroproliferation through the activation of TGFβ signaling, a well-established profibrotic pathway. It has been shown that TGFβ stimulates human and mouse fibroblasts to secrete EVs enriched in Program Death Ligand-1 (PD-L1) protein [72]. EV-associated PD-L1 further promotes lung fibroblast activation, proliferation, and paracrine-mediated suppression of T cell proliferation [72]. We and others have shown that EV-associated miRNAs from injured tissue have lower levels of antifibrotic TGFβ -regulating miRs [21]. In particular, miR-144-3p, miR-142-3p miR-34-b, and miR-503-5p are depleted in injured lung epithelial cell-derived EVs. MiR-144-3p and miR-142-3p also inhibit phosphorylation of the SMAD2 protein in a mouse bleomycin-induced lung injury model [21]. Moreover, macrophage-derived EVs downregulate the expression of TGFβ receptor 1 (TGFβ-R1) and profibrotic genes in lung epithelial cells and lung fibroblasts through miR-142-3p [73]. Others showed that IL-1β activation on lung fibroblasts stimulated EV-associated prostaglandin E2 (PGE2) production. PGE2 in the EVs can downregulate TGFβ signaling on lung fibroblasts through autocrine and paracrine effects [74]. In summary, EVs carry proteins and microRNAs that modulate TGFβ signaling, a major pathway of lung fibroproliferation.

Another major profibrotic pathway modulated by EVs is the Wnt signaling pathway. Lung-derived EVs following bleomycin injury are enriched in WNT5A, a noncanonical Wnt ligand [75]. Exposing healthy lung fibroblasts to WNT5A triggered fibroblast activation and proliferation, leading to collagen production [75]. Similarly, exposing lung fibroblasts to TGFβ also induced WNT5A and promoted fibroblast activation and fibrotic expansion. Another study showed that EVs derived from lung fibroblasts of IPF patients were enriched in fibronectin (FN), a protein that accumulates in lung fibrotic tissue and increases [20] fibroblast invasiveness and lung fibro proliferation through focal adhesion kinase (FAK) and Src family kinases activation [20]. Others have demonstrated that Wnt activation further induces FN expression [76].

Another principal driving pathway in lung fibrosis is cellular senescence [77,78]. The involvement of EVs in other chronic lung diseases such as COPD and lung cancer through cellular senescence was reviewed previously [79]. In ILD, one study showed that IPF fibroblast EV-associated miR-23b-3p and miR-494-3p induced human bronchial epithelial cells= senescence by suppressing SIRT3 expression, mitochondrial damage in epithelial cells, and senescence [80]. We have shown that EVs from human IPF lungs and bleomycin-injured mouse lungs had lower levels of antifibrotic miRNAs such as miR-144-3p, miR-142-3p, miR-34-b, and miR-503-5p that regulate cellular senescence, suggesting that EVs from diseased states are prosenescent and profibrotic [21].

In summary, EVs play a significant role in ILD progression and, in the diseased state, drive lung fibroproliferative processes by activating pathogenic pathways in healthy tissue (Figure 1). This suggests that EVs from patient samples can serve as diagnostic tools and that diseased tissue may be responsive to EV therapy. The former supports the role of EVs as biomarkers, and the latter supports their application in therapy [21,74].

## 4. Extracellular Vesicles as Biomarkers

While the roles of EVs as a diagnostic tool are rapidly developing for malignant neoplastic diseases such as lung cancer, this application remains in its early stages regarding ILD. The contributing factors include insufficient characterization of EV cargoes in patient samples [81,82] and the low efficiency of obtaining EVs from clinical specimens for an accurate diagnosis. However, in recent years, the tremendous progression of diagnostic EVs in cancer has helped facilitate the utilization of EVs as biomarkers for ILDs. Efforts to support the future translation of EV as biomarkers are focused on the development of more practical yet efficient methods of EV isolation from clinical specimens [83].

miRNAs are widely studied cargo in EVs and have long been implicated in ILD pathogenesis. Differential expression of circulating miRNAs during different stages of IPF progression suggests that these extracellular miRNAs may be used as biomarkers for diagnosis and staging [84]. We postulate that differential expression of these miRNAs may also be found in circulating EVs. One study demonstrated an increase of profibrotic miRNAs (miR-7 and miR-125) and decreased antifibrotic miRNA (miR-141) in serum-derived EVs from IPF patients, suggesting that serum EV-associated miRNAs play a significant role in pathogenesis [85]. In the preclinical model, circulating EV-miR-21-5p was first found in mice with bleomycin-lung injuries. This model was further validated in the human serum of IPF patients [86]. More importantly, serum EV-levels of miR-21-5p were correlated with increasing mortality risk at 30 months, supporting its role as a biological marker for disease severity. Recently, others demonstrated an upregulation of miR-21 in the serum-derived EVs of premature infants with chronic lung disease (CLD) indicating that miR-21 can be used as a predictive biomarker for CLD in this population [87]. Noncirculating EV-associated miRNAs are also evaluated; for example, one report showed that IPF patients displayed distinct miRNA profiles in sputum EVs compared to healthy controls, suggesting that these miRNAs may be profibrotic [88]. Further identification of miRNAs demonstrated that certain miRNAs (miR-142-3p, miR-33a-5p, and Let-7d-5p) were correlated with disease severity. Although the development and internal validation cohort in this study were small, it provides a foundation for future work in a larger cohort. The second study evaluating lung fibroblast-derived EVs revealed similar findings; IPF lung fibroblast-derived EVs were enriched in miR-23b-3p and miR-494-3p, which correlated with disease severity [80]. Together, microRNA-associated EVs derived from sputum, BALF, and circulation can potentially be used as diagnostic and prognostic markers for ILD.

EV protein cargo has also gained interest as a biomarker recently. Proteomic analysis of EVs from BALF of 15 pulmonary sarcoidosis patients demonstrated increased levels of pro-inflammatory proteins such as cell-cell glycoproteins, LPS-binding proteins, vitamin D binding proteins (VDBP), and complement activating proteins [68]. VDBP level was further validated in a different cohort of BALF- and serum-derived EVs of sarcoidosis patients, reinforcing the utility of EV protein biomarkers in patients with pulmonary sarcoidosis. In addition to EV-associated cargos, the role of cell-specific EVs was reported among ILD patients, whereby circulating endothelial cell-derived EVs were significantly elevated as quantitated by flow cytometry [89].

In conclusion, despite the need for further investigation findings from sputum-, BALF-, and serum-derived EVs represent promising points of ILD diagnostic development.

## 5. Therapeutic Roles of Extracellular Vesicles

EVs represent more versatile and scalable therapeutics than cell therapy as they are a noncell based therapeutic approach, and these include recapitulating the full therapeutic effect of the parent cells themselves, which are nonliving and unchanging entities, and are less immunogenic [36,37,83]. The reparative, antifibrotic, and immunomodulatory properties of EVs represent critical points of mechanistic interventions in ILD treatment [22]. At the current stage, therapeutic EVs studies are in the preclinical phase and many challenges remain to be investigated before EVs can be transitioned into clinical studies [33,37]. Nonetheless, results from preclinical studies, are encouraging and indicate promising therapeutic options (Table 1).

Many therapeutic cell-derived EV candidates with antifibrotic properties exist for ILDs, including mesenchymal stem cells (MSC), cardiosphere-derived cells, and healthy lung tissue [98]. The most common source of MSCs is bone marrow. Recent work demonstrated that bone marrow-MSC-derived EVs suppress fibroblast proliferation and lung fibrosis by downregulating Frizzled Class Receptor 6 (*FZD6*) expression in fibroblasts through miR-29b-3p both in vitro and in vivo [90]. Others also showed that MSC-EVs-miR-214-3p may alleviate radiation-induced lung injury and lung fibrosis in mice by downregulating ataxia telangiectasia mutated (ATM) kinase through suppression of ATM/P53/P21 signaling and thereby inhibiting the senescence in endothelial cells [91]. It has also been shown that human bone-MSC EVs can prevent and revert bleomycin-induced lung fibrosis [92] as demonstrated by the administration of these EVs at or after bleomycin injury. The protective effects of these EVs were more pronounced than their reversal of existing fibrosis. The EVs were also shown to modulate local macrophage- and monocyte-proinflammatory activation, though the specific mechanism remains to be described. EVs derived from MSC and lung spheroid cells (LSCs) also demonstrated therapeutic efficacy in bleomycin- and silica-induced lung fibrosis [93]. Using transcriptomic and proteomic analysis of EV cargo, it was postulated that miR-10a, miR-99, and let-7 along with MMP2 and TIMPs were mediating these beneficial effects. This study was among the first to administer EV therapy through inhalational administration and demonstrate antifibrotic activity.

Adipose mesenchymal stem cells (ADSCs) are another source of EVs that have been shown to significantly reduce primary lung injury and pulmonary fibrosis in rat lungs exposed to fine particulate matter less than 2.5 µm (PM2.5), demonstrating that pulmonary fibrosis is mitigated in part by let-7d-5p targeting TGFβ-R1 [94]. Specifically, the intratracheal administration of rat ADSC-EVs 1 h after PM2.5 exposure can suppress TGFβ-RI on alveolar type II cells that in turn prevents downstream events of the TGFβ signaling pathway.

Cardiosphere-derived cells (CDCs) are a population of cardiac-derived stromal cells with demonstrated disease-modifying activity in cardiac and skeletal muscle indications [95]. The mechanisms of action of CDC therapy include immunomodulation, antifibrosis, and tissue repair [95]. CDCs function primarily through the secretion of EVs which, on their own, recapitulate the full therapeutic effect of the parent cells [99]. Recent studies have identified CDC–EV cargo that mediates their immunomodulatory and antifibrotic effects. For instance, CDC–EVs contain keystone immunoregulatory miRs, including miR-146a and miR-181b, which attenuate innate immune inflammation through the suppression of the NFκB pathway and classical macrophage polarization, respectively [99]. MiR-146a also targets the TGFβ pathway through the suppression of Smad4. Furthermore, they contain other noncoding RNA species, including YRNA, which attenuate inflammation through the activation of IL10 secretion. Also, CDC–EVs are enriched in miR-24, which suppresses TGFβ signaling through the targeting of Furin [100]. Given what is already known about CDC–EV cargo and the pathways these signals affect in recipient cells, CDC–EVs represent promising therapeutics for ILDs.

Other cellular sources of therapeutic EVs are obtained directly from well-differentiated cells, including lung epithelial cells, lung fibroblasts, and immune cells. Our group has shown that lung-derived epithelial cell EVs, collected from syndecan-1 deficient mice, alleviated bleomycin-induced lung fibrosis by the activity of miRNA-associated EVs such as miR-142-3p, miR-144-3p, miR-34b, and miR-503-5p by targeting various profibrotic pathways [21]. Others have demonstrated that human amnion epithelial cell-derived EVs modulate bleomycin-induced lung remodeling and revert lung fibrosis by reducing lung epithelial cell damage and subepithelial myofibroblast accumulation [101]. EV-mediated lung repair was found to be superior to pirfenidone (a current antifibrotic agent for IPF) and its effects were enhanced by serelaxin, suggesting that these EVs may be deployed as adjunctive therapy. Additionally, human amnion epithelial cell-derived EVs exert their antifibrotic and regenerative effects in bleomycin lung injuries in both young and aged mice [96]. These effects are presumed to be associated with miRNAs and proteins targeting profibrotic signaling pathways.

A few cell lines have also been considered as sources of therapeutic EVs. For example, the neutrophil cell line (HL-60) secretes EVs containing miR-1343 that can be taken up by alveolar epithelial cells and lung fibroblasts, further inhibiting TGFβ, a major profibrotic regulatory signaling pathway modulator [97]. Macrophages are another source of EVs that contain miR-142-3p, which inhibits a profibrotic activation in epithelial cells and lung fibroblasts by downregulating TGFβRI [73]. The EVs from IL-1β activated human lung fibroblasts contained PGE2 that inhibited myofibroblast differentiation and collagen production, implicating their antifibrotic properties for in vivo testing [74].

Currently, there are nearly 600 clinical trials (registered, ongoing, and completed) worldwide using EVs in diagnostic and therapeutic capacities. Of these only 32 are related to lung disease. More than 60% of these studies focused on pulmonary malignancies, 20% focused on severe acute respiratory syndrome coronavirus 2 (SARS-CoV2) or COVID-19 infection, and the remainder was distributed among COPD, sarcoidosis, cystic fibrosis, and liver transplantation (www.clinicaltrials.gov with search terms “extracellular vesicles, microvesicles, exosomes”). Despite the sizable increase of clinical trials over the years, challenges with therapeutic translation of EVs remain. As with all other therapeutics, the primary challenge is effective delivery. Despite many features of EVs that make them an attractive therapeutic platform, they have little to no specific tissue tropism, as most biodistribution studies of systemically infused EVs demonstrate [102,103]. Secondly, owing to their diminutive size, retaining them at a certain site or target tissue is challenging as they are easily flushed by circulation. Therefore, for indications that require the direct application of EVs to localized areas (e.g., ischemic injury), patches of various hydrogels have been developed as scaffolds to provide consistent retention and release [104,105]. More specific EV delivery-enhancement efforts have focused on altering the EV surface for improved tissue retention, including the use of chemically-embedded targeting antibodies or homing peptides [106]. Furthermore, for EVs to be a viable therapy, a steady source of producer cells is needed. Current strategies are built around a single use paradigm for producer cells. For primary cell-derived EVs (currently the majority of therapeutic EVs), scale-up becomes prohibitive and would otherwise require a steady and ample source. One answer to this challenge has been the immortalization of producer cells as a steady source of therapeutic EVs [107]. More recently, we and others have developed immortalization strategies guided by the mechanistic understanding of the pathways that contribute to producer cell (and EV) potency [108,109]. Other strategies aimed at altering the EV surface include the stable introduction of a transgene expressing a chimeric protein comprising the intracellular domain of a conserved EV-surface receptor such as CD63 [110] or Lamp2b [111], which is fused to the extracellular portion of a protein of interest.

Furthermore, as the investigation of EV cargo advances into the identification of defined factors, new platforms, and strategies for enhancing EV bioactivity and improving therapeutic potency continue to be developed. Indeed, many reports of therapeutic EVs have identified therapeutically relevant cargo that triggers tissue healing and repair through attenuation of inflammation and fibrosis, promotion of tissue protection, repair, and angiogenesis [34,112]. Many of these defined factors include miRNAs, other small RNAs, and proteins [95,102]. For instance, some strategies exploit the ubiquitination pathway by tagging proteins of interest with various versions of a “ubiquitination” signal, which prompts efficient binding and EV loading by the trafficking protein Ndfip1 [111]. Another strategy for protein loading into EVs involves fusing the protein of interest with the trafficking domain of Nef1, an HIV lipid raft trafficking protein with the ability to sort proteins into vesicles [113]. For small RNAs, more straightforward methods that do not involve the engineering of the producer cell are the prevalent approach. These include the common nucleic acid and small molecule drug loading methods including EV electroporation, cationic lipid-mediated transfection of EVs or producer cells, EV sonication, and temperature–shock transfection, among others [114,115]. Recent findings, however, have elucidated the mechanisms by which miRs are preferentially sorted into EVs, including the identification of 3′ EV-localization sequences (GGAG/UGCA and AGG/UAG) that were recognized for sorting into EVs by heterogeneous ribonuclear proteins [116]. Therefore, an emerging strategy is the stable transfection of a miR of interest incorporating this trimer sequencing to increase packaging in the EV payload (Figure 2).

In summary, EVs show significant promise as a noncell regenerative medicine treatment [117]. Though challenges remain, the constant progress in mechanistic understanding, the identification of defined factors, and the development of genetic and material engineering are moving EVs towards clinical viability [37,118].

## 6. Conclusions

EVs have revolutionized our approach and understanding of ILD pathogenesis and can potentially be used as biomarkers and therapeutics. Although the literature on the basic biology and clinical applications of EVs remains wanting, increasing progress is being made in the preclinical setting. Our review highlights relevant studies that have deepened our understanding of these clinical implications and highlighted the limitations of EV applicability and knowledge gaps in critical areas of ILD research (Figure 3).

Biomarkers for ILDs are essential and would offer clinicians the ability to provide evidence-based personalized care to ILD patients. Currently, the application of ILD biomarkers is hindered by the complexity of ILD diagnostic criteria [81,82,119,120,121]. The main obstacles are the sensitivity and specificity of such biomarkers that require high case numbers for statistical validation. A few have been recently explored [81,122]. Uniquely, EVs can carry and preserve cargo, making them a promising target for research. The primary challenges are efficient and reproducible EV isolation protocols. EV isolation by ultracentrifugation followed by purification using a density gradient (UC-SDG) is the current gold standard for obtaining a pure EV preparation. However, this method involves significant loss and requires special equipment (ultracentrifuge), making it costly and impractical. The advancement in EV isolation techniques will certainly accelerate a translation to clinical research and their application as therapeutic agents. Furthermore, the availability of patient-derived EVs for biomarker mining remains extremely limited. Future high throughput analysis of circulating biomarkers in ILDs may include serum- and plasma-derived EVs that could be purified with high yields for comprehensive multi-omics research. Preclinical biomarkers identified from in vitro and in vivo animal models may not be directly applicable to human biology. A comprehensive database of human EV cargo contents from respiratory biological specimens such as BALF or serum is still not well-developed. The reference database is essential in understanding the global changes of cellular functions inside and outside the lung microenvironment, allowing a more accurate integration of disease processes.

The therapeutic promise of EVs as cell-free based therapy is upward trending with high potential as described earlier. Despite clear advantages, specific challenges require further basic research, process and product engineering and development. Undoubtedly, preclinical results have been promising. However, the current lack of standardization of EV isolation, purification, and characterization methods is a major challenge to the translation of therapeutic EVs into clinical studies. Standardization and harmonization of these protocols are vital for the accurate evaluation and demonstration of reproducibility needed for clinical translation [3]. Furthermore, little is known regarding EV kinetics and biodistribution in animal models, and, more so, in human subjects. EV labeling and tracking technologies still have significant limitations including quantifiable and reliable measures of EV tracking in vivo [123,124]. Finally, EV engineering to improve scalability, drug delivery, and therapeutic relevance and potency remain active areas of research and discovery.

In summary, EV research is fast evolving and offers significant promise in the area of ILD therapeutic research, allowing for the clinical utilization of EV in the near future. Effective tools of EV purification, cargo evaluation, tracking, and therapeutic cell sourcing represent the primary objectives for the continued translation of these next-generation regenerative medicine therapeutics.

## Figures and Tables

**Figure 1 diagnostics-11-00087-f001:**
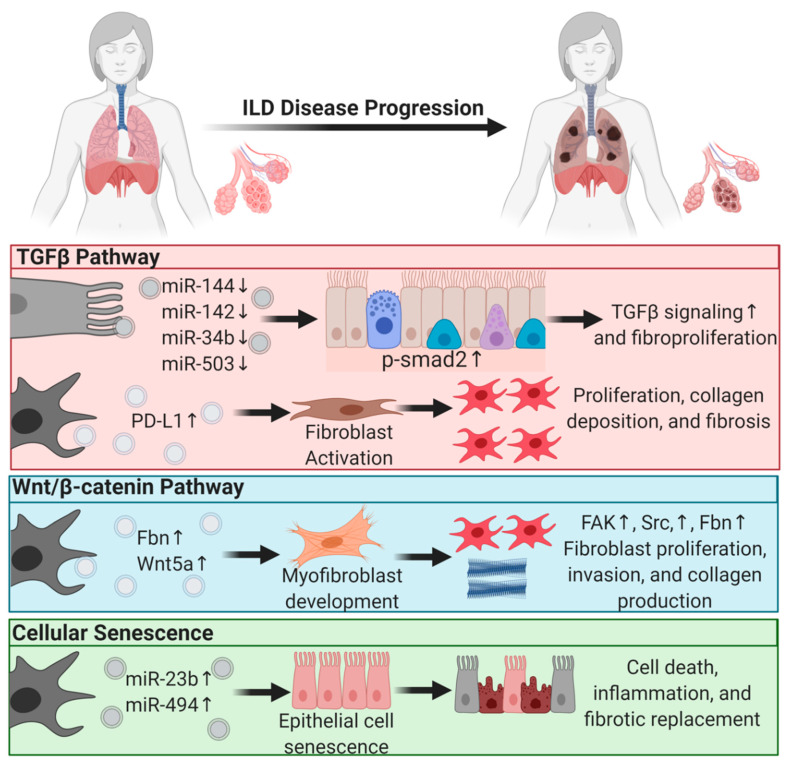
Extracellular vesicles regulate profibrotic signaling pathways in interstitial lung disease pathogenesis. Major profibrotic pathways are regulated by extracellular vesicles’ (EVs) cargo molecules in driving the principal pathways of lung fibroproliferative diseases. Arrows pointing up reflect upregulation and arrows pointing down signify downregulation.

**Figure 2 diagnostics-11-00087-f002:**
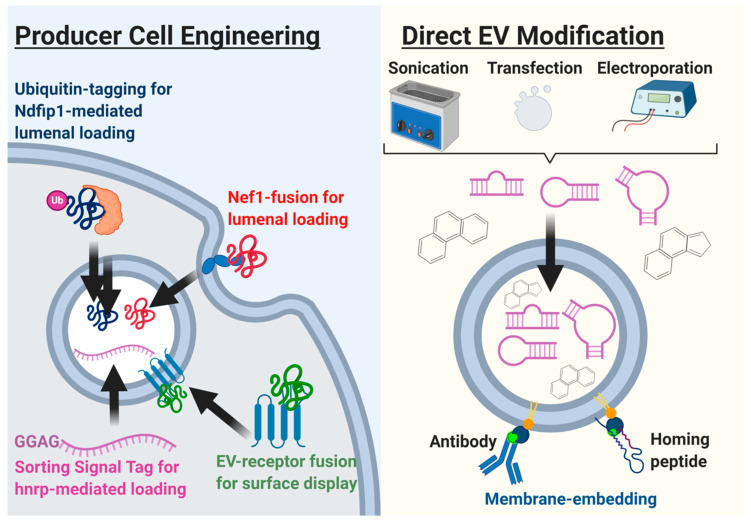
The schematic illustrates the engineering and production of extracellular vesicles.

**Figure 3 diagnostics-11-00087-f003:**
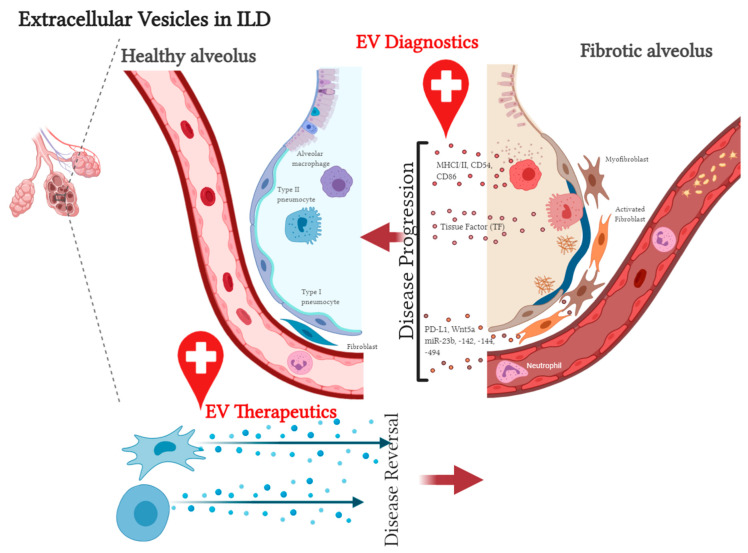
Extracellular vesicles in interstitial lung diseases. The secretion of pathologic EVs from the injured and fibrotic microenvironment, principally inflammatory cells, epithelial cells, activated fibroblasts, and myofibroblasts triggers further damage in local and distant healthy tissue. Identifying the signals carried by these EVs aids in the development of diagnostic tools to diagnose and track disease progression. EVs derived from therapeutic cell types can also be used to reverse disease phenotype through tissue immunomodulation and tissue repair.

**Table 1 diagnostics-11-00087-t001:** Preclinical therapeutic application of EVs in interstitial lung diseases and fibrosis.

Experimental Lung Fibrosis Model	EV Source	EV–Cargoes	Outcomes	Reference
Bleomycin	BMSC	miR-29-3p	Lung fibrosis *	[90]
Radiation	MSC	miR-214-3p	Lung injury, Lung fibrosis *	[91]
Bleomycin	Bone-MSC	unknown	Lung fibrosis *	[92]
Bleomycin, Silica	MSC	miR-10a, miR-99, let-7, MMP2, TIMP	Lung fibrosis *	[93]
PM < 2.5 µm	ADSC	let-7d-5p	Histology	[94]
Silica	ADSC	unknown	Lung fibrosis *, lung compliance	[95]
Bleomycin	Amnion epithelial cells	unknown	Lung fibrosis *	[96]
Bleomycin	Amnion epithelial cells	unknown	Lung fibrosis *, lung compliance	[97]

* Lung fibrosis was quantitated by hydroxyproline or collagen content, and/or Ashcroft score. BMSC = Bone marrow-derived mesenchymal stem cells; MSC = mesenchymal stem cells; MMP = metalloproteinase; TIMP = tissue inhibitor of metalloproteinase; PM = particle matters; ADSC = adipose mesenchymal stem cells.

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
