# Peer review of "Diagnostic and Therapeutic Applications of Extracellular Vesicles in Interstitial Lung Diseases"

_diagnostics, 2021, doi:10.3390/diagnostics11010087_

Round 1
Reviewer 1 Report
This manuscript is a review on the topic of extracellular vesicles in interstitial lung disease. The Authors present a thorough and comprehensive review of the available literature. The manuscript is well written and easy to follow. I find the topic of great interest and expect the paper, if published, to be of interest for many readers.
I only have two minor comments that I would like to share with the Authors.
- Page 1; lines 37-38. The statement that current treatments do not impact mortality might be inaccurate as some meta analyses have in fact shown a reduction in the relative risk of mortality in IPF patients treated wit antifibrotic drugs compared with placebo (see for example, Nathan SD et al., Lancet Respir Med 2017;5:33-41)
- On page 2, lines 68-69, the Authors seem to imply that while exosomes are capable of selective packaging of cargo, ectosomes are not. Although the differentiation of these vesicles is notoriously difficult, it has been shown that the composition of ectosomes/microparticles is different based on the stimulus used to generate them (see for example Jimenez JJ et al., Thromb Res 2003; 109:175-180).
Author Response
We greatly appreciate all constructive comments from the reviewers. Our responses are point-by-point below.
# 1. Page 1; lines 37-38. The statement that current treatments do not impact mortality might be inaccurate as some meta analyses have in fact shown a reduction in the relative risk of mortality in IPF patients treated wit antifibrotic drugs compared with placebo (see for example, Nathan SD et al., Lancet Respir Med 2017;5:33-41).
Response: The sentence is modified, and the reference provided by the reviewer is added (Ref #15). Specifically, “Although current treatments provide significant morbidity and mortality benefits [15], they are not curative.
# 2. On page 2, lines 68-69, the Authors seem to imply that while exosomes are capable of selective packaging of cargo, ectosomes are not. Although the differentiation of these vesicles is notoriously difficult, it has been shown that the composition of ectosomes/microparticles is different based on the stimulus used to generate them (see for example Jimenez JJ et al., Thromb Res 2003; 109:175-180).
Response: We have added 2 references (#38-39), including the reviewer’s reference by Jimenez et al (# 38), to further clarify the cargo packaging into ectosomes. The point raised by the reviewer touches on a central point in EV signaling. Several observations point to a non-stoichiometric representation of proteins and small RNAs in the cargo of exosomes compared to parent cells. Ectosomes including microvesicles, however, more closely match the cytoplasmic contents of the cells reflecting the passive nature of their loading. Indeed, ectosomes carry cell contaminants and debris including ER proteins, double-stranded DNA, ribosomes, and other internal cell structures suggesting that the loading is non-directed. EVs derived from the same cell type under different stimuli are notably different, however, whether this is due to the change of cell content itself in response to the stimuli (per current understanding) or if programmed loading also exists for ectosomes remains to be demonstrated.
Reviewer 2 Report
Ibrahim et al has discussed in this review article role of EVs as biomarkers for lung diseases as well as the potential use of EVs as therapy. The authors provide a description about ILDs, disease pathogenesis and role of EVs in disease progression. Role of EVs as therapeutics is also described. Review articles of similar nature have been published in the literature; therefore this manuscript does not offer new or novel information. Overall, the manuscript is well written and provides a succinct overview of the current state of EV research in the field of ILDs.
Minor Comments-
1.Page 2, Line 48-49: Provide reference. Also, EVs can regulate disease pathogenesis either directly or indirectly, please clarify.
- More detail on the EV engineering such as overexpression of candidate miRNA or proteins is suggested. Discuss challenges for therapy for lung disease such as EV routes of administration, dosing, EV scalability etc.
- An update on status of clinical trials for EVs in lung diseases is recommended.
- Efficacy studies for EVs in pre-clinical models in a table format is suggested
Author Response
We greatly appreciate the reviewer's comments.
Response: The additional schematic figure (new Figure 1) is added per the reviewer’s suggestion.
Reviewer 3 Report
This is a very interesting review on the role of EVs in interstitial lung diseases (ILD). While the concept and the collected data look promising, we would assume that not only specialists in lung diseases read this review. To make it more comprehensive, it will be helpful to briefly describe main factors/cell crosstalk/signaling pathways associated with pathogenenesis of ILD. This will explain the role/participance of EVs and their cargo in disease development. Also, it will be helpful to add some schemes demonstrating how EVs provide interactions between signaling pathway, to show schematic role of diagnostic tools, such as miRNa and illustrate how EVs can be used for therapeutic purposes. These schemes will make the text more crisp and understandable.
Author Response
We greatly appreciate the reviewer's suggestion. We have addressed the concerns as follows.
- Page 2, Line 48-49: Provide reference. Also, EVs can regulate disease pathogenesis either directly or indirectly, please clarify.
Response: We assume by indirectly you mean through diagnostics and directly pertains to their utility as therapeutic candidates. We have clarified that in the portion of the review indicated.
- More detail on the EV engineering such as overexpression of candidate miRNA or proteins is suggested. Discuss challenges for therapy for lung disease such as EV routes of administration, dosing, EV scalability, etc.
Response: An in-depth discussion of EV engineering techniques, particularly surface expression and luminal cargo is discussed beginning on lines 289-326 on pages 8-9. A summary figure (new Figure 2) is also provided.
- An update on the status of clinical trials for EVs in lung diseases is recommended.
Response: A summary of clinical trials is provided in the revised version of the review starting from line 283-288 on page 8.
- Efficacy studies for EVs in pre-clinical models in a table format is suggested
Response: The new Table 1 is added.